# The Distinction between Dematiaceous Molds and Non-Dematiaceous Fungi in Clinical and Spiked Samples Treated with Hydrogen Peroxide Using Direct Fluorescence Microscopy

**DOI:** 10.3390/jof9020227

**Published:** 2023-02-09

**Authors:** Elchanan Juravel, Itzhack Polacheck, Batya Isaacson, Arie Dagan, Maya Korem

**Affiliations:** 1Department of Internal Medicine, Hadassah Medical Center and Faculty of Medicine, Hebrew University of Jerusalem, Jerusalem 9112101, Israel; 2Department of Clinical Microbiology and Infectious Diseases, Hadassah Medical Center and Faculty of Medicine, Hebrew University of Jerusalem, Jerusalem 9112101, Israel; 3The Concern Foundation Laboratories at the Lautenberg Center for Immunology and Cancer Research, The Hebrew University Medical School, IMRIC, Jerusalem 9112102, Israel; 4Department of Developmental Biology and Cancer Research, Institute for Medical Research Israel-Canada, The Hebrew University Medical School, Jerusalem 9112102, Israel

**Keywords:** dematiaceous molds, hydrogen peroxide, melanin, fluorescence microscopy, black fungi

## Abstract

Dematiaceous fungi are pigmented molds with a high content of melanin in their cell walls that can cause fatal infections in immunocompromised hosts. Direct microscopy is the main method for the rapid diagnosis of dematiaceous fungi in clinical specimens. However, it is often difficult to distinguish their hyphae from non-dematiaceous hyphae and yeast pseudohyphae. Our aim was to develop a fluorescence staining method that targets melanin for the detection of dematiaceous molds in clinical specimens. Glass slide smears of clinical samples and sterile bronchoalveolar lavage spiked with dematiaceous and non-dematiaceous fungi were treated with hydrogen peroxide, and digital images were recorded using direct microscopy with different fluorescent filters. The images of fungi were compared for their fluorescence intensity using the NIS-Elements software. The fluorescent signal between dematiaceous and non-dematiaceous fungi demonstrated a markedly increased mean intensity for dematiaceous molds following hydrogen peroxide treatment (7510.3 ± 10,427.6 vs. 0.3 ± 3.1, respectively, *p* < 0.0001). No fluorescent signal was detected in the absence of hydrogen peroxide. “Staining” fungal clinical specimens with hydrogen peroxide, followed by fluorescence microscopy examination, can differentiate between dematiaceous and non-dematiaceous fungi. This finding can be used for the detection of dematiaceous molds in clinical specimens and enables the early and appropriate treatment of infections.

## 1. Introduction

Dematiaceous fungi are pigmented septate hyphae molds, distributed worldwide in soil or plants [1], that are associated with a variety of clinical syndromes, including life-threatening disseminated infections (phaeohyphomycosis) in both immunocompromised and immunocompetent individuals [1,2,3]. Therefore, a timely diagnosis is highly important. A feature common to dematiaceous fungi is the presence of dihydroxyphenylalanine (DHN)-derived melanin in their cell walls, which is responsible for the characteristic dark color of their conidia and hyphae [2]. Melanin has been demonstrated in smaller quantities in many non-dematiaceous fungi, including *Histoplasma capsulatum*, *Paracoccidioides brasiliensis*, *Aspergillus* spp., and *Cryptococcus neoformans* [1,4,5,6], and is considered a major virulence factor acting as a potent free radical scavenger, protecting against the oxidants generated through immune effector cells, such as those resulting from the oxidative burst of phagocytes [7].

Despite various testing methodologies available in the modern clinical laboratory, culture remains the mainstay of diagnosis of dematiaceous fungi [3,8], though the turnaround time might be long [7]. Therefore, there is great value in initial diagnosis through direct microscopy of tissue specimens using potassium hydroxide (KOH) and calcofluor white staining [4]. However, direct microscopy is often inaccurate in distinguishing dematiaceous mold hyphae from pseudohyphae of yeasts, due to their similar morphology of irregularly swollen hyphae [2], and from non-dematiaceous hyphae (e.g., *Aspergillus* spp.), mainly following antifungal treatment that can alter hyphal morphology. This may lead to incorrect diagnosis and treatment, especially if the fungus fails to grow later on agar culture. Consequently, there is a need for a method that can differentiate dematiaceous fungi from other fungi without awaiting cultures.

Melanin of the human retinal pigment epithelium (RPE) is thought to react to oxidative stress. In previous studies, oxidative stress to the RPE was induced by hydrogen peroxide and measured using a vital fluorescent probe in fluorescence microscopy [9,10]. Further studies established the fluorescence feature of oxidized melanin as common to all natural and synthetic melanins as well as melanin-like pigments because it could be induced in eumelanin, pheomelanin, and neuromelanin, both in vivo and in vitro [10,11,12]. In addition, the kinetics of melanin destruction were recorded based on the accumulation of fluorescent low-molecular-weight reaction products detected at an excitation wavelength of 440–470 nm [9].

On the basis of these studies, we hypothesized that high-content melanin in the cell walls of dematiaceous fungi may produce a fluorescent response after the addition of hydrogen peroxide to a smear. The aim of this study was to evaluate a method for the detection of dematiaceous fungi in clinical specimens treated with hydrogen peroxide by examining the fluorescent response in direct microscopy and comparing it with the response observed with non-dematiaceous fungi.

## 2. Materials and Methods

### 2.1. Isolates

The clinical isolates in this study were from a collection stored at −80 °C at the Hadassah Medical Center Microbiology Laboratory. *Alternaria alternata* (NRRL 54028), *Aspergillus fumigatus* (NRRL 2427), *Aspergillus flavus* (NRRL 3518), *Aspergillus terreus* (NRRL 269), *Candida parapsilosis* (ATCC 22019), and *Candida albicans* (ATCC 90028) were included as reference organisms in all laboratory tests (Table 1).

Stored clinical isolates were thawed, inoculated on Sabouraud Dextrose Agar (SDA, Novamed, Jerusalem, Israel), incubated for 3–5 days at 30 °C and then re-identified through microscopic evaluation and matrix-assisted laser desorption/ionization time-of-flight mass spectrometry (MALDI-TOF MS) (bioMerieux, Marcy-l ’Etoile, France).

### 2.2. Smear Preparation

For the preparation of the smears, we used (1) clinical samples obtained from hospitalized patients, as per clinical indication, for mycological workup and found positive for fungal elements in direct microscopy using the calcofluor white 0.1% stain with subsequent fungal growth in culture (*n* = 3, Table 1); (2) bronchoalveolar lavage (BAL) fluid, which was obtained from hospitalized patients who underwent bronchoscopy for various reasons and spiked with fungal strains from the laboratory collection (*n* = 18, Table 1). All BAL specimens were without fungal elements prior to spiking, according to direct fluorescence microscopy with calcofluor white 0.1% and SDA cultures. Briefly, 3 mL of each BAL specimen was spiked with one strain of a yeast or a mold with an adjusted inoculum size of 1 × 10^6^–5 × 10^6^ spores/mL via microscopic enumeration with a hemocytometer, as previously described [13]. Table 1 describes the strains used in this study and their source. 

### 2.3. Fluorescence Microscopy Staining

First, 20 µL of spiked BAL fluid, two swabs containing fungal isolates, and one nasal specimen were applied to clean glass slides and allowed to air-dry. The following smears were prepared for each sample: (1) Control smears included (a) no stains or additives and (b) 30 µL calcofluor white 0.1% (Sigma, St. Louis, MI, USA) mixed with 10% KOH reagent (Merck, Darmstadt, Germany), as formerly described; (2) test smears included (a) 20 µL of 10% hydrogen peroxide and 20 µL of potassium carbonate (1M) stain and (b) 20 µL of 30% hydrogen peroxide and 20 µL of potassium carbonate (1M) stain. Potassium carbonate was added to enhance oxidation by hydrogen peroxide since a strong base ionizes the phenolic hydroxyl groups of melanin [14].

Following staining, the cover-slipped smears were kept at room temperature for three hours and then examined using a fluorescent microscope (Zeiss Axioscope, Jena, Germany) with an OLYMPUS DP72 camera.

### 2.4. Microscopic Examination

The smears with no additives or stains (control smears “1a” above) were examined under a bright field and two different fluorescent filters as described below, for the identification of fungal elements and for comparison with hydrogen-peroxide-stained smears. Microscopy images were recorded in two different magnifications (×200, ×400).

The calcofluor-white-stained smears were examined under a fluorescent microscope in two different magnifications (×200, ×400), and the images were recorded. 

The smears stained with hydrogen peroxide were examined in a bright field and in two different fluorescent filters: Filter 1—barrier (LP) 520 nm, excitation 450–490 nm, and emission 510 nm; Filter 2—barrier (LP) 420 nm, excitation 365 nm, and emission 395 nm. The images obtained with a bright field and each of the fluorescent filters were recorded at two different magnifications (×200 and ×400). The transition between the filters was performed with no movement of the examined slide to ensure that the same field in each smear was examined and photographed in a bright field and in fluorescence mode. 

All smears were prepared as duplicates for self-comparison and laboratory control. Information about the prepared smears and the images that were recorded can be found in Appendix A. A schematic description of the process of smear preparation, microscopic evaluation, and image recording can be found in Figure 1.

### 2.5. Image Recording

The recorded images were evaluated for fluorescence using the imaging software NIS-Elements (Nikon Instruments Inc., Tokyo, Japan), which measures the area in pixels of the obtained fluorescent signals (binary area). The defined range of fluorescence intensity is 0–255. The intensity threshold was set at 120 according to a control image of fungal elements with no stain. Thus, an area in pixels with fluorescence intensity above 120 was calculated using the software and is presented in this paper.

### 2.6. Statistical Analysis

Statistical analysis was performed using GraphPad Prism (version 9.5.0). Groups were compared against each other, and a two-tailed Mann–Whitney (non-parametric) test or Student’s *t*-test (parametric) was used. A cutoff value to discriminate dematiaceous from non-dematiaceous fungi was determined by the receiver operating characteristic (ROC) curve with a comparison of fluorescent signal values (30% hydrogen peroxide, Filter 1, and ×200 magnification). The cutoff value was set at a value above which both sensitivity and specificity reached 100%. *p* < 0.05 was considered significant in all studies.

## 3. Results

Twenty-one isolates were included in the study: eight dematiaceous molds (including one reference organism), seven *Aspergillus* species (including three reference organisms), four *Candida* species (including two reference organisms), one species from the Mucorales order and one *Fusarium* species.

A total of 252 images were recorded. A comparison of the mean fluorescent signal between dematiaceous and non-dematiaceous fungi, calculated for combined filters, magnifications, and hydrogen peroxide concentrations, demonstrated a markedly increased intensity of dematiaceous molds (7510.3 ± 10,427.6 vs. 0.3 ± 3.1, respectively, *p* < 0.0001), with the highest intensities received for an *Alternaria* spp. and the lowest for *Exserohilum rostratum* (24,594 vs. 1003, respectively; averages of all measurements at both concentrations of hydrogen peroxide) (Figure 1, Appendix A). No fluorescent signal was detected in the absence of hydrogen peroxide (0 ± 0 vs. 0 ± 0, *p* > 0.9999). Differentiation according to the hydrogen peroxide concentration added to the dematiaceous mold smears revealed higher fluorescence intensity when 30% hydrogen peroxide was added, though the difference did not reach statistical significance (10,435.5± 13,706.5 for 30% vs. 4585.1 ± 3929.7 for 10% hydrogen peroxide, *p* = 0.1239). There was no difference in intensities obtained for either filter used (1 vs. 2) (8305.3 ± 9897.1 vs. 6715.3 ± 11,032.7, *p* = 0.55) or magnification (×200 vs. ×400) (7036 ± 10,288 vs. 7984 ± 10,707, respectively, *p* = 0.577), though higher signals were observed when ×400 magnification was used in a few dematiaceous molds (Appendix A). A fluorescent signal cutoff level of 295 was found to predict the presence of dematiaceous molds, with a sensitivity and specificity of 100% (0.67 < 95%CI < 1, *p* < 0.05). Fluorescence microscopy images of *Alternaria alternata* representing the dematiaceous fungi group are presented in Figure 2. 

## 4. Discussion

In this study, we showed that the fluorescence intensity received for the images of specimens containing dematiaceous fungi following the addition of hydrogen peroxide was significantly higher than non-dematiaceous fungi. All dematiaceous fungi significantly surpassed the defined intensity threshold, and the only non-dematiaceous fungi (*Aspergillus fumigatus*) that reached the threshold was 30-fold lower in intensity than *E. rostratum* that had the lowest fluorescence intensity among the dematiaceous group (Appendix A). A fluorescence intensity cutoff of 295 was sensitive and specific to discriminate dematiaceous from non-dematiaceous fungi. 

The common feature of dematiaceous fungi is the presence of melanin in their cell walls [2], which acts as a potent free radical scavenger and protects the fungus against oxidative stress [4]. Melanin’s potential fluorescent reaction with hydrogen peroxide, a powerful oxidative stressor, was formerly demonstrated in a study that recorded the kinetics of melanin destruction based on the accumulation of fluorescent low-molecular-weight reaction products following the addition of hydrogen peroxide [8]. Melanin in fungi is derived primarily from either dihydroxyphenylalanine (L-DOPA) or dihydroxynaphthalene (DHN). Dematiaceous fungi contain only DHN melanin [4]. As other studies established the fluorescence of oxidized melanin to be a property of oxidized melanin as opposed to a contamination byproduct [10], and that this feature is common to all natural and synthetic melanins [11], it can be concluded that the results of our study can be attributed to the higher concentration of melanin in the cell walls of dematiaceous fungi. 

An evaluation of the hydrogen peroxide concentration required to achieve a fluorescent response showed that a concentration of 30% resulted in two-fold higher fluorescence intensity and may be preferable to a concentration of 10%, although this difference did not reach statistical significance. This finding corresponds with other studies that showed increased fluorescent response with higher concentrations of hydrogen peroxide and longer time of exposure [9,12]. In our study, the time of exposure was set to approximately 3 h. Further research is needed to determine the required time of hydrogen peroxide exposure for optimal differentiation of dematiaceous molds from other fungi. There was no difference in the fluorescent signals obtained for dematiaceous fungi in ×200 and ×400 magnifications, though in a few, we observed a higher signal with ×400, which can be explained by different radiation exposure of molecules or exposure time.

The *dematiaceous* fungus *E. rostratum* had the lowest fluorescence intensity following the addition of hydrogen peroxide to a specimen smear. This could be related to the source of the specimen, as it was derived from a nasal biopsy, as opposed to most other dematiaceous molds, derived from a laboratory collection and spiked into BAL fluids. It is possible that a lower density of *E. rostratum* in the specimen smear led to decreased fluorescence intensity, and therefore, it is important to conduct a study with clinical specimens, as the expression of melanin by fungi may be related to growth conditions and fungal concentrations. 

These findings provide a simple method to distinguish dematiaceous molds from other fungi with the use of direct fluorescence microscopy and can be used for the early diagnosis of phaeohyphomycosis, with a potential impact on treatment decision making, especially in immunocompromised hosts [2]. The method can be potentially implemented when swollen septate hyphae are observed in calcofluor-white-stained clinical specimens in conventional fluorescent microscopy, particularly when the distinction between yeasts’ pseudohyphae and dematiaceous molds is uncertain. The method is inexpensive, with an estimated cost of laboratory hardware and consumables of US dollars ~1 per sample, in laboratories equipped with a fluorescence microscope and a dedicated camera.

There are several limitations to this study. Although there was an effort to mimic in vivo conditions by spiking BALs with fungal strains, only 3/21 of specimens were truly clinical specimens. This may affect the expression of melanin by fungi or their morphology (hyphae or pseudohyphae formation) [15]. Additionally, since BAL fluid was spiked with a constant inoculum (1–5 × 10^6^ spores/mL), the limit of detection of the hydrogen peroxide method was not determined in comparison with the conventional calcofluor white stain in fluorescence microscopy. We also did not examine the effect of hydrogen peroxide on the fluorescence of immature dematiaceous fungi that have a low content of cell wall melanin. Therefore, it is possible that the method we describe lacks sensitivity in low-inoculum models or early stages of infection. Finally, we did not prove that the fluorescent response was due to a reaction with melanin, although there is a strong basis to assume that this was indeed the case. 

## 5. Conclusions

In this study, we demonstrated that the addition of hydrogen peroxide to fungus-containing specimens could differentiate between dematiaceous and non-dematiaceous fungi, most probably due to a higher content of melanin in dematiaceous fungal cell walls that react with hydrogen peroxide with a consequent fluorescent response. This phenomenon can be used as a method for the detection of dematiaceous molds in clinical specimens and will facilitate the early and appropriate treatment of invasive fungal infections.

## Data Availability

Data supporting reported results can be found in the manuscript and in Appendix A.

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
