# Peer review of "The Distinction between Dematiaceous Molds and Non-Dematiaceous Fungi in Clinical and Spiked Samples Treated with Hydrogen Peroxide Using Direct Fluorescence Microscopy"

_jof, 2023, doi:10.3390/jof9020227_

Round 1

Reviewer 1 Report

The article “Distinction between dematiaceous molds and non-dematiaceous fungi in clinical and spiked samples treated with hydrogen peroxide using direct fluorescence microscopy” details a novel method based on fluorescence microscopy to differentiate between dematiaceous and non-dematiaceous fungi.

The paper is well written and contains interesting information for rapid differentiation of dematiaceous fungi submitted for routine identification.

It would be interesting to provide information of the cost of this methodology per sample.

Besides, it would be interesting for the readers if the author would propose different scenarios for the implementation of this novel methodology in low-income countries or in laboratories with rapid methods, such as MALDI-TOF, are available.

Reviewer 2 Report

1.     The aim of the study was to evaluate a method for early detection of dematiaceous fungi in clinical specimens. For early detection, the authors should have studied the various inoculum sizes of spores and should have demonstrated that their method can detect low inoculum size when compared with the conventional method. But authors have not studied the limit of detection of dematiaceous fungi spores which is detected by the specimens treated with hydrogen peroxide and comparing the same with the conventional fluorescence microscopy with Calcofluor White. This is one of the limitations of the study which can be added in the limitation part. Here authors demonstrated specimens containing dematiaceous fungi which can be detected by fluorescence microscopy if the specimen is treated by hydrogen peroxide and can be differentiated easily with the hyaline fungi. Therefore, it is suggested to remove “early” word from the aim of the study and same should be corrected in the whole manuscript.

2.     Authors have also not studied the immature dematiaceous fungi when the melanin content is low in the cell wall and possess difficulty in identification of dematiaceous fungi. It is another limitation of the study which may be added in the limitation part.

3.     It is suggested to conclude the cuff fluorescent signal for the dematiaceous fungi.

4.     It is suggested to add and compare mean fluorescent signal recorded in two different magnification – X200 and X400 for the dematiaceous fungi. It is also observed that fluorescent signal recorded for X400 were higher in some cases when compared to X200 magnification for the same field while it was lower in other cases. Similar finding was also observed by the authors for Aspergillus fumigatus NRRL 2427 in BAL. Authors may add this important finding with explanation.  

5.     References should be cited as per journal’s instruction – issue number may not be needed.
